# AUTOSAMPLING: SEARCH FOR EFFECTIVE DATA SAMPLING SCHEDULES

## ABSTRACT

Data sampling acts as a pivotal role in training deep learning models. However, an effective sampling schedule is difficult to learn due to its inherent high-dimension as a hyper-parameter. In this paper, we propose the AutoSampling method to automatically learn sampling schedules for model training, which consists of the multi-exploitation step aiming for optimal local sampling schedules and the exploration step for the ideal sampling distribution. More specifically, we achieve sampling schedule search with shortened exploitation cycle to provide enough supervision. In addition, we periodically estimate the sampling distribution from the learned sampling schedules and perturb it to search in the distribution space. The combination of two searches allows us to learn a robust sampling schedule. We apply our AutoSampling method to a variety of image classification tasks illustrating the effectiveness of the proposed method.

## 1 INTRODUCTION

Data sampling policies can greatly influence the performance of model training in computer vision tasks, and therefore finding robust sampling policies can be important. Handcrafted rules, e.g. data resampling, reweighting, and importance sampling, promote better model performance by adjusting the training data frequency and order (Estabrooks et al., 2004; Weiss et al., 2007; Bengio et al., 2009; Johnson & Guestrin, 2018; Katharopoulos & Fleuret, 2018; Shrivastava et al., 2016; Jesson et al., 2017). Handcrafted rules heavily rely on the assumption over the dataset and cannot adapt well to datasets with their own characteristics. To handle this issue, learning-based methods (Li et al., 2019; Jiang et al., 2017; Fan et al., 2017) were designed to automatically reweight or select training data utilizing meta-learning techniques or a policy network.

However existing learning-based sampling methods still rely on human priors as proxies to optimize sampling policies, which may fail in practice. Such priors often include assumptions on policy network design for data selection (Fan et al., 2017), or dataset conditions like noisiness (Li et al., 2019; Loshchilov & Hutter, 2015) or imbalance (Wang et al., 2019). These approaches take images features, losses, importance or their representations as inputs and use the policy network or other learning approaches with small amount of parameters for estimating the sampling probability. However, for example, images with similar visual features can be redundant in training, but their losses or features fed into the policy network are more likely to be close, causing the same probability to be sampled for redundant samples if we rely on aforementioned priors. Therefore, we propose to directly optimize the sampling schedule itself so that no prior knowledge is required for the dataset. Specifically, the sampling schedule refers to order by which data are selected for the entire training course. In this way, we only rely on data themselves to determine the optimal sampling schedule without any prior.

Directly optimizing a sampling schedule is challenging due to its inherent high dimension. For example, for the ImageNet classification dataset (Deng et al., 2009) with around one million samples, the dimension of parameters would be in the same order. While popular approaches such as deep reinforcement learning (Cubuk et al., 2018; Zhang et al., 2020), Bayesian optimization (Snoek et al., 2015), population-based training (Jaderberg et al., 2017) or simple random search (Bergstra & Bengio, 2012) have already been utilized to tune low-dimensional hyper-parameters like augmentation schedules, their applications in directly finding good sampling schedules remain unexploited. For instance, the dimension of a data augmentation policy is generally only in dozens, and it needs thousands of training runs (Cubuk et al., 2018) to sample enough rewards to find an optimal augmentation

policy because high-quality rewards require many epochs of training to obtain. As such, optimizing a sampling schedule may require orders of magnitude more rewards than data augmentation to gather and hence training runs, which result in prohibitively slow convergence.

To overcome the aforementioned challenge, we propose a data sampling policy search framework, named AutoSampling, to sufficiently learn an optimal sampling schedule in a population-based training fashion (Jaderberg et al., 2017). Unlike previous methods, which focus on collecting long-term rewards and updating hyper-parameters or agents offline, our AutoSampling method collects rewards online with a shortened collection cycle but without priors. Specifically, the AutoSampling collects rewards within several training iterations, tens or hundred times shorter than that in existing works (Ho et al., 2019; Cubuk et al., 2018). In this manner, we provide the search process with much more frequent feedback to ensure sufficient optimization of the sampling schedule. Each time when a few training iterations pass, we collect the reward from the previous several iterations, accumulate them and later update the sampling distribution using the rewards. Then we perturb the sampling distribution to search in distribution space, and use it to generate new mini-batches for later iterations, which are recorded into the output sampling schedule. As illustrated in Sec. 4.1, shortened collection cycles with less interference also can better reflect the training value of each data.

Our contributions are as follows:

- To our best knowledge, we are the first to propose to directly learn a robust sampling schedule from the data themselves without any human prior or condition on the dataset.
- We propose the AutoSampling method to handle the optimization difficulty due to the high dimension of sampling schedules, and efficiently learn a robust sampling schedule through shortened reward collection cycle and online update of the sampling schedule.

Comprehensive experiments on CIFAR-10/100 and ImageNet datasets (Krizhevsky, 2009; Deng et al., 2009) with different networks show that the Autosampling can increase the top-1 accuracy by up to 2.85% on CIFAR-10, 2.19% on CIFAR-100, and 2.83% on ImageNet.

## 2 BACKGROUND

### 2.1 RELATED WORK

Data sampling is of great significance to deep learning, and has been extensively studied. Approaches with human-designed rules take pre-defined heuristic rules to modify the frequency and order by which training data is presented. In particular, one intuitive method is to resample or reweight data according to their frequencies, difficulties or importance in training (Estabrooks et al., 2004; Weiss et al., 2007; Drummond et al., 2003; Bengio et al., 2009; Lin et al., 2017; Shrivastava et al., 2016; Loshchilov & Hutter, 2015; Wang et al., 2019; Johnson & Guestrin, 2018; Katharopoulos & Fleuret, 2018; Byrd & Lipton, 2018; Jesson et al., 2017). These methods have been widely used in imbalanced training or hard mining problems. However, they are often restricted to certain tasks and datasets based on which they are proposed, and their ability to generalize to a broader range of tasks with different data distribution may be limited. In another word, these methods often implicitly assume certain conditions on the dataset, such as cleanness or imbalance. In addition, learning-based methods have been proposed for finding suitable sampling schemes automatically. Methods using meta-learning or reinforcement learning are also utilized to automatically select or reweight data during training (Li et al., 2019; Jiang et al., 2017; Ren et al., 2018; Fan et al., 2017), but they are only tested on small-scale or noisy datasets. Whether or not they can generalize over tasks of other datasets still remain untested. In this work, we directly study the data sampling without any prior, and we also investigate its wide generalization ability across different datasets such as CIFAR-10, CIFAR-100 and ImageNet using many typical networks.

As for hyper-parameter tuning, popular approaches such as deep reinforcement learning (Cubuk et al., 2018; Zhang et al., 2020), Bayesian optimization (Snoek et al., 2015) or simply random search (Bergstra & Bengio, 2012) have already been utilized to tune low-dimensional hyper-parameters and proven to be effective. Nevertheless, they have not been adopted to find good sampling schedule due to its inherent high dimensiona. Some recent works tackle the challenge of optimizing high-dimensional hyper-parameter. MacKay et al. (2019) uses structured best-response functions and Jonathan Lorraine (2019) achieve this goal through the combinations of the implicit function theorem and efficient inverse Hessian approximations. However, they have not been tested on the task of optimizing sampling schedules, which is the major focus of our work in this paper.

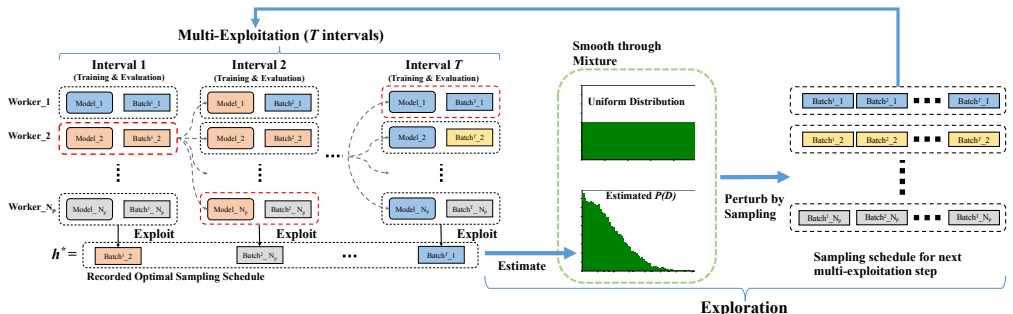

Figure 1: Overview of AutoSampling illustrated through one multi-exploitation-and-exploration cycle. a) The multi-exploitation step, illustrated by the left half, is the process of learning optimal sampling schedule locally. The same color of model for each worker indicates that the same model weight is cloned into it. Also for simplicity, in this figure we adopt the exploitation interval of length 1. b) The exploration step, shown by the right half, is to search in the sampling distribution space. Specifically, we estimate the sampling distribution from the schedules collected in the multi-exploitation step and perturb it to generate new sampling schedules for all workers.

## 2.2 POPULATION BASED TRAINING

Hyper-parameter tuning task can be framed as a bi-level optimization problem with the following objective function,

$$\min_{h \in \mathcal{H}} \mathcal{L}(\theta^*, h)$$
$$\text{subject to } \theta^* = \arg\max_{\theta \in \Theta} \text{eval}(\theta, h) \tag{1}$$

where $\theta$ represents the model weight and $h = (h_1, h_2, \cdots, h_T)$ is the hyper-parameter schedule for $T$ training intervals. Population based training (PBT) (Jaderberg et al., 2017) solves the bi-level optimization problem by training a population $\mathcal{P}$ of child models in parallel with different hyper-parameter schedules initialized:

$$\mathcal{P} = \{(\theta_i, h_i, t)\}_{i=1}^{N_p} \tag{2}$$

where $\theta_i, h_i$ respectively represents the child model weight, the corresponding hyper-parameter schedule for the training interval $t$ on worker $i$, and $N_p$ is the number of workers. PBT proceeds in intervals, which usually consists of several epochs of training. During the interval, the population of models are trained in parallel to finish the lower-level optimization of weights $\theta_i$.

Between intervals, an exploit-and-explore procedure is adopted to conduct the upper-level optimization of the hyper-parameter schedule. In particular for interval $t$, to exploit we evaluate child models on a held-out validation dataset:

$$h_t^*, \theta_t^* = \arg\max_{p_i = (\theta_i, h_i, t) \in \mathcal{P}} \text{eval}(\theta_i, h_i)$$
$$\theta^* \to \theta_i, i = 1, \cdots, N_p \tag{3}$$

We record the best performing hyper-parameter setting $h_t^*$ and broadcast the top-performing model $\theta_t^*$ to all workers. To explore, we initialize new hyper-parameter schedules for interval $t + 1$ with different random seeds on all workers, which can be viewed as a search in the hyper-parameter space. The next exploit-and-explore cycle will then be continued. In the end, the top-performing hyper-parameter schedule $h^* = (h_1^*, h_2^*, \cdots, h_T^*)$ can be obtained.

PBT is applied to tune low-dimenisal hyper-parameters such as data augmentation schedules (Ho et al., 2019; Jaderberg et al., 2017). However, it cannot be directly used for finding sampling strategies due to the high dimension. Unlike PBT, our AutoSampling adopts a multi-exploitation-and-exploration structure, leading to much shorter reward collection cycles that contribute to much more and effective rewards for sufficient optimization within a practical computational budget.

## 3 AUTOSAMPLING WITH SEARCHING

The overview of our AutoSampling is illustrated in Fig.1. AutoSampling alternately runs multi-exploitation step and exploration step. In the exploration step, we 1) update the sampling distribution

---

**Algorithm 1:** The Multi-Exploitation Step

---

**Input:** Training dataset $D$, population $\mathcal{P} = \{(\theta_i, h_i, t)\}_{i=1}^{N_p}$, number of workers $N_p$, number of exploitation intervals $T$, exploitation interval length $N_s$

Initialize $\mathbf{H}^* \leftarrow ()$

**for** $t = 1$ **to** $T$ **do**

    **for** $j = 1$ **to** $N_s$ **do**

        **for** $(\theta_i, h_{t,i}, t) \in \mathcal{P}$ **do**

            $\theta_i \leftarrow \triangledown \mathcal{L}(\theta_i, h_{t,i})$      $\triangleright$ update the weight of child model i

        **end for**

        $h_t^*, \theta_t^* = \arg\max_{\mathcal{P}} \mathrm{eval}(\theta_i, h_i)$

        $\mathbf{H}^* \leftarrow \mathbf{H}^* + h_t^*$        $\triangleright$ update the sampling for child model i

        **for** $i = 1$ **to** $N_p$ **do**

            $\theta_i \leftarrow \theta_t^*$             $\triangleright$ clone the optimal weight

        **end for**

    **end for**

**end for**

**Return** $\mathbf{H}^*, \mathcal{P}$

---

using the rewards collected from the multi-exploitation step (the sampling distribution is uniform distribution initially); 2) perturb the updated sampling distribution for child models so that different child models have different sampling distributions; 3) use the corresponding perturbed sampling distribution for each child model to sample mini-batches of training data. In the multi-exploitation step, we 1) train multiple child models using the mini-batches sampled from the exploration step; 2) collect short-term rewards from the child models. AutoSampling finishes with a recorded top-performing sampling schedule, which can be transferred to other models.

## 3.1 MULTI-EXPLOITATION BY SEARCHING IN THE DATA SPACE

In the multi-exploitation step, we aim to search locally in the data space by collecting short-term rewards and sub-schedules. Specifically, we wish to learn a sampling schedule for $T$ exploitation intervals. In each interval, there are a population $\mathcal{P}$ of $N_p$ child models. Denote $\mathbf{h}_{t,i}$ as the training data sub-schedule in the $t^{th}$ interval for the $i^{th}$ child model. When all of the $T$ exploitation intervals for the $i^{th}$ child model are considered, we have $\mathbf{H}_i = \{\mathbf{h}_{t,i} | t = 1, \ldots, T\} = \{x_1, \cdots, x_N\}$, where $N$ is the number of training data for the multi-exploitation step. Each interval consists of $N_s$ training iterations that is also equivalent to $N_s$ training mini-batches, where $N_s$ is the length of the interval. AutoSampling is expected to produce a sequence of training samples, denoted by $\mathbf{H}^*$, so that a given model is optimally trained. The population $\{\mathbf{H}_i\}$ forms the local search space, from which we aim to search for an optimal sampling schedule $\mathbf{H}^*$.

Given the population $\mathcal{P}$, we train them in parallel on $N_p$ workers. Once an interval of data $\mathbf{h}_{t,i}$ containing $N_s$ training batches have been used for training, we evaluate all child models and use the top evaluation performance as the reward. According to the reward, we record the top-performing weight and sub-schedule for the current interval $t$, in particular,

$$h_t^*, \theta_t^* = \arg\max_{p_i = (\theta_i, h_i, t) \in \mathcal{P}} \mathrm{eval}(\theta_i, h_{t,i}) \tag{4}$$

On the other hand, we update all child model weights of $\mathcal{P}$ by cloning into them with the top-performing weight $\theta_t^*$ so we can continue searching based on the more promising child. We will continue the exploit steps through the whole training process, and output the recorded optimal sampling schedule $\mathbf{H}^* = \{h_1^*, h_2^*, \cdots, h_T^*\}$. By using exploitation interval of mini-batches rather than epochs or even entire training runs adopted by earlier methods, AutoSampling may yield a better and more robust sampling schedule. It should be pointed out that even though in AutoSampling rewards are collected within a much shorter interval, they remain effective. As we directly optimize the sampling schedule, we are concerned with only the data themselves. The short-term rewards reflect the training value of data from the exploitation interval they are collected. But for global hyper-parameters such as augmentation schedules, short-term rewards may lead to inferior performance as these hyper-parameters are concerned with the overall training outcome. We describe the multi-exploitation with details in Alg.1.

---

**Algorithm 2:** Search based AutoSampling

---

**Input:** Training dataset $D$, population size $N_p$
Initialize $H^* \leftarrow ()$ , $P(D) \leftarrow$ uniform$(D)$ and initialize child models $\theta_1, \cdots, \theta_{N_p}$
**while** not end of training **do**
   **for** $i = 1$ **to** $N_p$ **do**
      Sample $h_i$ from Mixture$(\log(P(D) + \beta), N_u \times$ uniform$(D))$
   **end for**
   Initialize $\mathcal{P} = \{(\theta_i, h_i, t)\}_{i=1}^{N_p}$
   $\mathbf{H}^*, \mathcal{P} \leftarrow$ Alg.1
   Estimate $P(D)$ according to Equation (5)
   Update $P(D)$ according to Equation (6)
   $H^* \leftarrow H^* + \mathbf{H}^*$
**end while**
**Return** $H^*, P(D)$

---

## 3.2 EXPLORATION BY SEARCHING IN SAMPLING DISTRIBUTION SPACE

In the exploration, we search in sampling distribution space by updating and perturbing the sampling distribution. We first estimate the underlying sampling distribution $P(D)$ from the top sampling schedule $h^*$ produced in the multi-exploitation, that is, for $x \in D$,

$$P(x) = \frac{count(x \in \mathbf{H}^*)}{\sum_{x \in D} count(x \in \mathbf{H}^*)} \tag{5}$$

where $count(x \in \mathbf{H}^*)$ denotes the number of $x$'s appearances in $\mathbf{H}^*$. We further perturb the $P(D)$ and generate the sampling schedules on each worker for the later multi-exploitation. We introduce perturbations into the generated schedules by simply sampling from the multinomial distribution $P(D)$ using different random seeds. However, in our experiments, we observe that the distribution produced by $P(D)$ tends to be extremely skewed and a majority of the data actually have zero frequencies. Such skewness causes highly imbalanced training mini-batches, and therefore destabilizes subsequent model training.

**Distribution Smoothing** To tackle the above issue, we first smooth $P(D)$ through the logarithmic function, and then apply a probability mixture with uniform distributions. In particular for the dataset $D$,

$$P'(D) = Mixture(\log(P(D) + \beta), N_u \times \text{uniform}(D)) \tag{6}$$

where $\beta \geq 1$ is the smoothing factor and $N_u \times$ uniform$(D)$ denotes $N_u$ uniform multinomial distributions on the dataset $D$. The smoothing through the $\log$ function can greatly reduce the skewness, however, $\log(P(D) + \beta)$ may still contain zero probabilities for some training data, resulting in unstable training. Therefore, we further smooth it through a probability mixture with $N_u$ uniform distribution uniform$(D)$ to ensure presence of all data. This is equivalent to combining $N_u$ epochs of training data to the training batches sampled from $P(D)$, and shuffling the union. Once we have new diverse sampling schedules for the population, we proceed to the next multi-exploitation step.

We continue this alternation between multi-exploitation and exploration steps until the end of training. Note that to generate sampling schedule for the first multi-exploitation run, we initialize $P(D)$ to be an uniform multinomial distribution. In the end, we output a sequence of optimal sampling schedules $H^* = (\mathbf{H}_1^*, \cdots, \mathbf{H}_n^*)$ for $n$ alternations. The entire process is illustrated in details in Alg.2.

## 4 EXPERIMENTS

In this section, we present comprehensive experiments on various datasets to illustrate the performance of AutoSampling, and also demonstrate the process of progressively learning better sampling distribution.

### 4.1 ABLATION STUDY

For this part, we gradually build up and test components of AutoSampling on CIFAR-100, and then examine their performances on CIFAR-10 and ImageNet datasets. The training implementation details and computational complexity can be found in Appendix A.1.

Table 1: Performance on CIFAR-100 using different configurations of AutoSampling and baselines. Worker is the number of workers used and Interval is the exploitation interval in terms of batches.

| NETWORK | WORKER | INTERVAL | EXPLORATION TYPE | TOP1(%) |
|---|---|---|---|---|
| RESNET18 (ZHANG ET AL., 2019) | - | - | - | 78.34±0.05 |
| RESNET18 | 1 | - | UNIFORM | 78.46±0.035 |
| RESNET18 | 20 | 80 BATCHES | RANDOM | 78.76±0.003 |
| RESNET18 | 20 | 20 BATCHES | RANDOM | 78.99±0.003 |
| RESNET18 | 80 | 20 BATCHES | RANDOM | 79.09±0.017 |
| RESNET18 | 20 | 20 BATCHES | MIXTURE | **79.44**±0.020 |
| RESNET50 (JIN ET AL., 2019) | - | - | - | 79.34 |
| RESNET50 | 1 | - | UNIFORM | 79.70±0.023 |
| RESNET50 | 20 | 80 BATCHES | RANDOM | 80.55±0.129 |
| RESNET50 | 20 | 20 BATCHES | RANDOM | 81.05±0.064 |
| RESNET50 | 80 | 20 BATCHES | RANDOM | 81.19±0.072 |
| RESNET50 | 20 | 20 BATCHES | MIXTURE | **81.53**±0.088 |
| DENSENET121 | 1 | - | UNIFORM | 80.13±0.028 |
| DENSENET121 | 20 | 80 BATCHES | RANDOM | 80.62±0.694 |
| DENSENET121 | 20 | 20 BATCHES | RANDOM | **81.11**±0.127 |
| DENSENET121 | 80 | 20 BATCHES | RANDOM | 81.08±0.021 |
| DENSENET121 | 20 | 20 BATCHES | MIXTURE | 80.97±0.006 |

Table 2: Experiments on CIFAR-10.

| NETWORK | EXPLORATION TYPE | TOP1(%) |
|---|---|---|
| RESNET18 | UNIFORM | 93.01±0.009 |
| RESNET18 | RANDOM | **95.86**±0.003 |
| RESNET18 | MIXTURE | 95.80±0.018 |
| RESNET50 | UNIFORM | 93.60±0.004 |
| RESNET50 | RANDOM | **96.10**±0.002 |
| RESNET50 | MIXTURE | 96.09±0.070 |

Table 3: Experiments on ImageNet.

| NETWORK | EXPLORATION TYPE | TOP1(%) |
|---|---|---|
| RESNET18 | UNIFORM | 70.38 |
| RESNET18 | RANDOM | 72.07 |
| RESNET18 | MIXTURE | **72.91** |
| RESNET34 | UNIFORM | 74.09 |
| RESNET34 | RANDOM | 76.11 |
| RESNET34 | MIXTURE | **76.92** |

**Adding Workers** To look into the influence of the worker numbers, we conduct experiments using worker numbers of 1, 20, 80 respectively with the same setting ($N_s = 20$ with random exploration). With the worker number of 1, the experiment is simply the normal model training using stochastic gradient descent. To show the competitiveness of our baselines, we also include state-of-the-art results on CIFAR-100 with ResNet-18 and ResNet-50 (Zhang et al., 2019; Jin et al., 2019). We notice significant performance gain using the worker number of 20 for ResNet-18, ResNet-50 and DenseNet-121 (He et al., 2015; Huang et al., 2017), as illustrated in Table 1. However, we note that increasing worker number from 20 to 80 only brings marginal performance gains across various model structures, as shown in Table 1. Therefore, we set the worker number to be 20 for the rest of the experiments.

**Shortening Exploitation Intervals** To study the effects of the shortened exploitation interval, we run experiments using different exploitation intervals of 20 and 80 batches(iterations) respectively. As shown in Table 1, models with the shorter exploitation interval of 20 batches(iterations) perform better than the one with the longer exploitation interval across all three network structures, conforming to our assumptions that the reward collected reflects value of each data used in the exploitation interval. This result adheres to our intuition that shorter exploitation interval can encourage the sampler to accumulate more rewards to learn better sampling schedules. For the rest of this section we keep the exploitation interval of 20.

**Adding Exploration Type** We further add mixture as the exploration type to see the effects of learning the underlying sampling distribution, and completing the proposed method. As shown in Table 1, with ResNet-18 and ResNet-50 we push performance higher with the mixture exploration, and outperform the baseline method by about 1 and 1.8 percentage on CIFAR-100 respectively. However, we found that it is not true in the case of DenseNet-121 and this case may be attributed to the bigger capacity of DenseNet-121.

**Generalization Over Datasets** In addition, we experiment on other datasets. We report the results on CIFAR10 in Table 2 and the results of ResNet-18, ResNet-34 on ImageNet in Table 3. For CIFAR-10, we notice that the mixture and random exploration methods are comparable while both outperforming the uniform baseline, and we believe it is due to the simplicity of the dataset. In the more challenging

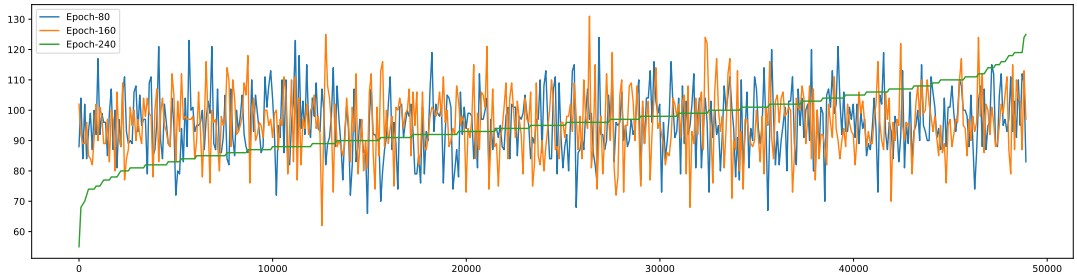

Figure 2: The comparison between histograms estimated from the sampling schedules of Epoch 80, 160 and 240 from CIFAR-100 with ResNet-18. We divide the 50000 training images into 500 segments of 100 images, and calculate the histograms of total data counts of all segments. We reorder the $x$-axis based on the ranking of data counts for epoch 240 for easier comparison.

Table 4: Static vs dynamic sampling schedule on CIFAR-100 (%)

| NETWORK | SAMPLING TYPE | | |
|---------|---------|--------|---------|
| | UNIFORM | STATIC | DYNAMIC |
| RESNET18 | 78.46±0.035 | 78.80±0.007 | **79.44**±0.020 |
| RESNET50 | 79.70±0.023 | 80.21±0.014 | **81.53**±0.088 |

ImageNet, the mixture exploration outperforms the random exploration by a clear margin. We also compare our AutoSampling with some recent non-uniform sampling methods on CIFAR-100, which can be found in Appendix A.2.

## 4.2 STATIC VS DYNAMIC SCHEDULES

We aim to see if the final sampling distribution estimated by our AutoSampling is sufficient to produce robust sampling schedules. In another word, we wish to know training with the AutoSampling is either a process of learning a robust sampling distribution, or a process of dynamically adjusting the sampling schedule for optimal training. To this end, we conduct training using different sampling schedules. First, we calculate the sampling distribution estimated throughout the learning steps of AutoSampling, and use it to generate the sampling schedule of a full training process, which we denote as STATIC. Moreover, we denote the sampling schedule learned using AutoSampling as DYNAMIC, since AutoSampling dynamically adjust the sampling schedule alongside the training process. Finally, we denote the baseline method as UNIFORM, which uses the sampling schedule generated from uniform distribution.

We report results on CIFAR-100 with ResNet-18 and ResNet-50 in Table 4. Model trained with STATIC sampling schedules exceeds the baseline UNIFORM significantly, indicating the superiority of the learned sampling distribution over the uniform distribution. It shows the ability of AutoSampling to learn good sampling distribution. Nonetheless, note that models trained with DYNAMIC sampling schedules outperform models trained with STATIC, by a margin bigger than the one between STATIC and UNIFORM. This result shows the fact that despite the AutoSampling's capability of learning good sampling distribution, its flexibility during training matters even more. Moreover, this phenomenon also indicates that models at different stages of learning process may require different sampling distributions to achieve optimal training. One single sampling distribution, even gradually estimated using AutoSampling, seems incapable of covering the needs from different learning stages. We plot the histograms of data counts in training estimated from schedules of different learning stages with ResNet-18 on CIFAR-100 in Fig.2, showing the great differences between optimized sampling distributions from different epochs.

## 4.3 ANALYZING SAMPLING SCHEDULES LEARNED BY AUTOSAMPLING

To further investigate the sampling schedule learned by AutoSampling, we review the images at the tail and head part of the sampling spectrum. In particular, given a sampling schedule learned we rank all images based on their appearances in training. Training images at the top and bottom of the order are extracted, corresponding to high and low probabilities of being sampled respectively. In Fig.3, we show 4 classes of exemplary images. The images of low probability tend to have clearer imagery

Figure 3: Example images on the head and tail of the sampling spectrum. The images on the left are the ones with low sampling probability, while the images on the right more likely to be sampled. We obtain these images using AutoSampling with the ResNet-18 model on CIFAR-100.

Table 5: Transfer of sampling distributions learned by three model structures to ResNet-50 on CIFAR-100 (%). UNIFORM denotes the baseline result using uniform sampling distribution.

| NETWORK | SAMPLING SCHEDULE SOURCE | | | |
|---|---|---|---|---|
| | UNIFORM | RESNET18 | RESNET50 | DENSENET121 |
| RESNET50 | 79.70±0.023 | 80.27±0.014 | 80.21±0.014 | 80.47±0.194 |

features enabling easy recognition, while the images of high probability tend to be more obscure, indicating that the sampling schedule may show hard samples mining effects. However, as shown in A.3 and Fig. 4, the loss values and probabilities of being sampled seem to be not highly correlated, which indicates more potential of AutoSampling beyond visually hard example mining. In addition, we notice the images of low probability also contain low quality images. For instance, in Fig.3 the leftmost image of CAMAL class contains only legs. This shows that AutoSampling may potentially rule out problematic training data for better training.

Furthermore, we examine the transfer ability of sampling distributions learned by AutoSampling to other network structures. Specifically, we run training on ResNet-50 (He et al., 2015) using STATIC sampling schedule generated by three distributions learned by AutoSampling on 3 different models. As shown in Table 5, using sampling schedules learned by AutoSampling from other models, we demonstrate similar improvements over the UNIFORM baseline. This result, combined with the above observations on images of different sampling probability, indicates that there may exist a common optimal sampling schedule determined by the intrinsic property of the data rather than the model being optimized. Our AutoSampling is an effort to gradually converge to such an optimal schedule.

## 4.4 DISCUSSIONS

The experimental results and observations from Section 4.2 and 4.3 shed light on the possible existence of an optimal sampling schedule, which relies only on the intrinsic property of the data and the learning stage of the model, regardless of the specific model structure or any prior knowledge. The learned sampling schedule may provide enough rewards in the searching process, leading to sufficient convergence compared to other related works. Once obtained, the optimal sampling schedule may also be generalized over other model structures for robust training. Although AutoSampling requires relatively large amount of computing resources to find a robust sampler, we want to point out that the efficiency of our method can be improved through better training techniques. Moreover, the possibility of an optimal sampling schedule relying solely on the data themselves may indicate more efficient sampling policy search algorithms, if one can quickly and effectively determine data value based on its property.

## 5 CONCLUSIONS

In this paper, we introduce a new search based AutoSampling scheme to overcome the issue of insufficient rewards for optimizing high-dimensional sampling hyper-parameter by utilizing a shorter period of reward collection. We use a shortened exploitation interval to search in the local data space and provide sufficient rewards. For the exploration step, we estimate sampling distribution from the searched sampling schedule and perturb it to search in the distribution space. We test our method and it consistently outperforms the baseline methods across different benchmarks.

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

## A  APPENDIX

### A.1  IMPLEMENTATION DETAILS

**Experiments on CIFAR** We use the same training configuration for both CIFAR-100 and CIFAR-10 datasets, which both consist of 50000 training images. In particular, for model training we use the base learning rate of 0.1 and a step decay learning rate schedule where the learning rate is divided by 10 after each 60 epochs. We run the experiments for 240 epochs. In addition, we set the training batch size to be 128 per worker, and each worker is for one Nvidia V100 GPU card.

We run the explore step for each $N_u + 1$ epochs with $N_u = 3$, but note that we take the first explore step after the initial 20 epochs to better accumulate enough rewards. The experiments require 4800 epochs of training for 20 workers, and roughly 14 hours of training time.

**Experiments on ImageNet** For ImageNet which consists of 1.28 million training images, we adopted the base learning rate of 0.2 and a cosine decay learning rate schedule. We run the experiments with 100 epochs of training. For each worker we utilize eight Nvidia V100 GPU cards and a total batch size of 512. Eight workers are used for all ImageNet experiments, and the rest of the setting adheres to that of CIFAR experiments. In addition, we utilize FP16 computation to achieve faster training, which has almost no drop in accuracy in practice. The experiments require 800 epochs of training for 8 workers, and roughly 4 days of training time.

### A.2  COMPARISON WITH EXISTING SAMPLING METHODS

To better illustrate the effectiveness of our AutoSampling method, we conduct experiments in comparison with recent non-uniform sampling methods DLIS (Johnson & Guestrin, 2018) and RAIS (Katharopoulos & Fleuret, 2018). DLIS (Johnson & Guestrin, 2018) achieves faster convergence by selecting data reducing gradient norm variance, while RAIS (Katharopoulos & Fleuret, 2018) does so through approximating the ideal sampling distribution using robust optimization. The comparison is recorded in Table 6.

First, we run AutoSampling using Wide Resnet-28-2 (Zagoruyko & Komodakis, 2016) on CIFAR-100 with the training setting aligned roughly to (Katharopoulos & Fleuret, 2018). AutoSampling achievs improvement of roughly 3 percentage points ($73.37\pm1.09\% \rightarrow 76.24\pm1.02\%$), while Katharopoulos & Fleuret shows improvement of 2 percentage points ($66.0\% \rightarrow 68.0\%$). Second, we report the comparison between AutoSampling and RAIS on CIFAR-100. Johnson & Guestrin shows no improvement ($76.4\% \rightarrow 76.4\%$) on accuracy and 0.027 ($0.989 \rightarrow 0.962$) decrease in validation loss, while our method shows improvement of 0.008 ($78.6\% \rightarrow 79.4\%$) on accuracy and 0.014 ($0.886 \rightarrow 0.872$) decrease in validation loss. As such, our method demonstrates significant improvements over existing non-uniform sampling methods.

Table 6: Comparisons between AutoSampling and existing sampling methods on CIFAR-100

| Methods | Network | Baseline (%) | With method (%) | Improvement (%) |
|---------|---------|--------------|-----------------|-----------------|
| DLIS | WRN-28-2 | 66.0 | 68.0 | 2.0 |
| AutoSampling (ours) | WRN-28-2 | 73.37±1.09 | 76.24±1.02 | **2.87** |
| RAIS | ResNet18 | 76.4 | 76.4 | 0.0 |
| AutoSampling (ours) | ResNet18 | 78.46±0.035 | 79.44±0.020 | **0.98** |

### A.3  COMPARISON BETWEEN LEARNED SAMPLING SCHEDULES AND DATA LOSS VALUES

To further interpret the learned sampling schedules, we compare the sampling frequency of each training image and its loss values in different epochs during training of CIFAR-100 with ResNet-18. We draw the comparison for randomly selected 500 training images in Fig. 4 for epoch 80, 160, and 240. As shown in the figure, across different learning stages, the correlation between loss values and sampling frequencies of training data is not obvious. The high chance of being sampled by AutoSampling does not necessarily lead to high loss values, which demonstrates that AutoSampling

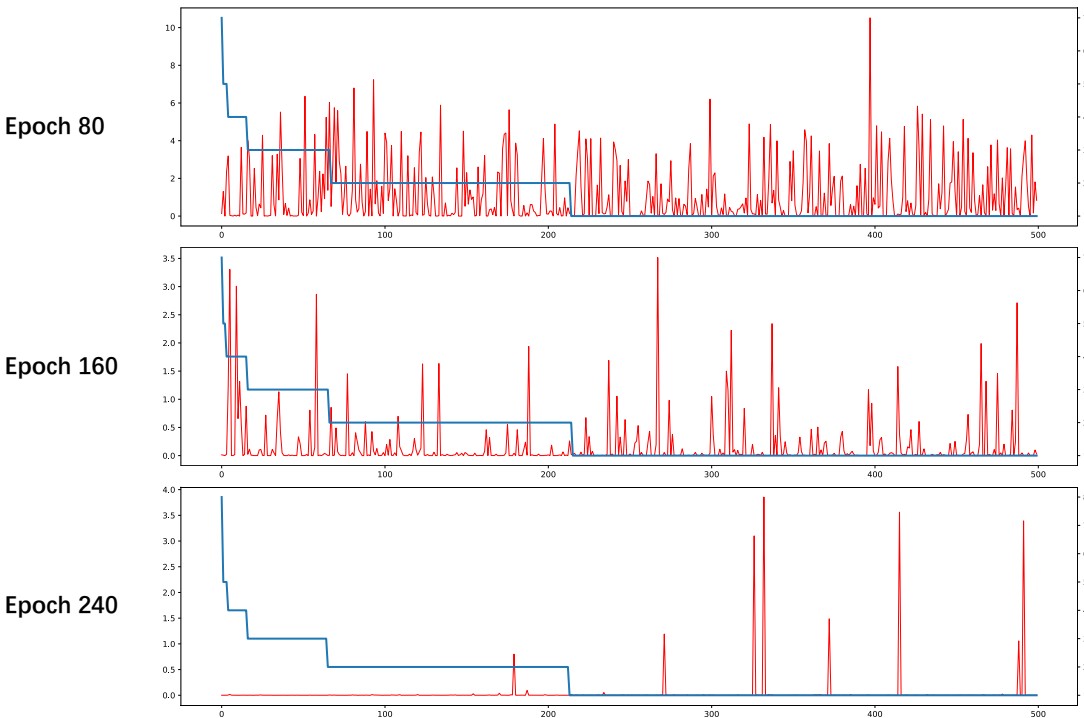

Figure 4: The comparison between the sampling frequency of each training image and its loss values of Epoch 80, 160 and 240 from CIFAR-100 with ResNet-18. We randomly selected 500 training images, and calculate their sampling frequency and loss values. The x-axis is the indexes of 500 training images, while the left y-axis denotes loss values and the right y-axis denotes the sampling frequency. The blue line represents the sampling frequencies and the red lines represents the loss values of all 500 images. As we can see from the figure, the two lines are not obviously correlated.

is not merely over-sampling difficult samples as pointed by the loss. The resulting sampling schedule learned by AutoSampling would be significantly different from the one guided by loss. Moreover as the training progresses the loss values of data are reduced, which is expected.

