# OpenReview forum: "AUTOSAMPLING: SEARCH FOR EFFECTIVE DATA SAMPLING SCHEDULES"
_ICLR.cc/2021/Conference — Reject_

### Official Review · AnonReviewer1 · 2020-10-30
**Unclear novelty and method, as well as inconclusive experimental results.**

**Rating:** 3
**Confidence:** 5

**Review:**

This paper discusses an approach to perform importance sampling by reviewing performance over past batches and suggesting future batches.

First I will comment on the listed contributions:

> • To our best knowledge, we are the first to propose to directly learn a robust sampling
schedule from the data themselves without any human prior or condition on the dataset.

From this description I believe this paper has done it before: "CASED: Curriculum Adaptive Sampling for Extreme Data Imbalance".

The paper seems to go beyond this and focuses on multi node training so this may be a way to refocus the paper in contrast to the existing work.

> • We propose the AutoSampling method to handle the optimization difficulty due to the high
dimension of sampling schedules, and efficiently learn a robust sampling schedule through
shortened reward collection cycle and online update of the sampling schedule.

It is not clear to me what the intuition of P(x) is. If H* is the "optimal sampling schedule" and then P is defined to be the ratio of x in H* over all x in H* then it is not clear to me what you are converging to. It would be better for the reader to make this as easy to understand as possible. Also, I think this would be the place where this approach is different from the CASED approach.

On the experiments to demonstrate that this method is better: The replicate runs seem to almost have no variance. It is the case that only the model init was varied between runs with fixed train/valid/test sets? If so this does not seem sufficient to confirm that this method works because the data you sample from is always the same. This should be randomized at least for CIFAR.

Typo: "Algorithm 2: Search based AuoSampling"

---

> ### Author Response · Authors · 2020-11-23
> **Response to Reviewer 1**
>
> We thank the reviewer for the valuable feedback and would like to address some of the reviewer’s concerns.
>
> * "From this description I believe this paper has done it before: "CASED: Curriculum Adaptive Sampling for Extreme Data Imbalance". The paper seems to go beyond this and focuses on multi node training so this may be a way to refocus the paper in contrast to the existing work."
>     * We will include CASED in the related works and would like to discuss and clarify the significant differences between the proposed method and CASED.  As we understand, CASED is a data re-sampling framework dedicated to train robust segmentation models for highly imbalanced datasets, in particular for lung nodule detection. We would like to point out two fundamental differences between two methods. Firstly, the AutoSampling method is a general method which by its formulation does not rely on any prior or condition of the datasets, while CASED assumes the dataset to be highly imbalanced and its formulation depends on the lung nodule detection itself. Secondly, the goal of AutoSampling is to find an optimal sampling schedule capable of transferring among different models, while CASED is intended to train robust segmentation models for lung nodule detection. The potential generalizability and purpose of the two methods are quite different to our understanding.
>
> * "It is not clear to me what the intuition of P(x) is. If H* is the "optimal sampling schedule" and then P is defined to be the ratio of x in H* over all x in H* then it is not clear to me what you are converging to."
>     * We wish to further clarify on our notations. $P(x)$ as defined in Equation 5 in Section 3.2 is the sampling distribution estimated from the recorded optimal sampling schedule $\mathbf{H}^*$ collected from the last multi-exploitation step. Both $P(x)$ (or $P(D)$) and $\mathbf{H}^*$  denotes the current local optimum, which are recorded and used to further search in the sampling distribution space. What we are interested in finding is the recorded optimal sampling schedules $\mathbf{H}^*$  from all the multi-exploitation steps, as defined as in the last sentence of Section 3.2.
>
> * "The replicate runs seem to almost have no variance. It is the case that only the model init was varied between runs with fixed train/valid/test sets? If so this does not seem sufficient to confirm that this method works because the data you sample from is always the same. This should be randomized at least for CIFAR."
>     * We do run the CIFAR-10/100 experiments with randomized model weight initializations and sampling schedules. We do this to rule out the fluctuations on performance, which is a common practice for running experiments on small-scale datasets in computer vision and AutoML literatures [1,2]. It is also a common practice in computer vision literatures to run experiments with fixed train/validation/test datasets for the fair comparison between concurrent works.
>
> Reference
> 1. Daniel Ho, Eric Liang, Ion Stoica, Pieter Abbeel, and Xi Chen. Population based augmentation: Efficient learning of augmentation policy schedules. CoRR, abs/1905.05393, 2019. URL http: //arxiv.org/abs/1905.05393.
> 2. Ekin Dogus Cubuk, Barret Zoph, Dandelion Mane ́, Vijay Vasudevan, and Quoc V. Le. Autoaugment: Learning augmentation policies from data. CoRR, abs/1805.09501, 2018. URL http://arxiv. org/abs/1805.09501.

---

### Official Review · AnonReviewer3 · 2020-10-31
**A paper marginally above average**

**Rating:** 6
**Confidence:** 4

**Review:**

The authors mainly concentrate on data sampling. To address the issue of optimizing high-dimensional sampling hyper-parameter in data sampling and release the requirement of prior knowledge from current methods, the authors introduce a searching-based method named AutoSampling. This method is comprised of exploration step and exploitation step which are conducted alternatively. The exploitation step train multi child models with current sampling strategy and save the best model for next iteration. while the exploration step estimates the sampling distribution according to the sampled data in exploitation step and rectifies it to sample all data possibly. The authors have conducted sufficient experiments to verify the superior of their method, especially for the effectiveness and generalizability.

Advantages:
l	The exploitation step and exploration step in AutoSampling is interesting, it is straightforward that this method can work well as the sampling strategy is updated dynamically according to the current state of model.
l	The proposed AutoSampling is simple and effective, one can implement it without much effort.
l	This method owns great generalizability and does not require any knowledge prior.
l	This paper is well organized and written.

Disadvantages:
l	In Table 1, the number of workers does have an influence on performance, and this influence is positively correlated in my opinion, however, we can see a performance degradation for DenseNet121. The authors did not explain this.
l	The transferability of the gained optimal sampling schedule is discussed in Section 4.4, a simple experiment is recommended.

---

> ### Author Response · Authors · 2020-11-23
> **Response to Reviewer 3**
>
> We thank the reviewer for the valuable feedback and positive comments, and would like to address some of the reviewer’s concerns.
>
> *  "In Table 1, the number of workers does have an influence on performance, and this influence is positively correlated in my opinion, however, we can see a performance degradation for DenseNet121."
>     * This is a great question. Our experiments on ResNet models show the positive correlation between the number of workers and performance; however, there are other factors at play, such as the batch size which may influence the quality of rewards. Therefore, one reason we think may account for the performance degradation for DenseNet 121 is that the number of workers along is not a deciding factor in terms of performance enhancement, other hyper-parameters such batch size may also need to be adjusted.
>
> * "The transferability of the gained optimal sampling schedule is discussed in Section 4.4, a simple experiment is recommended."
>     * We conducted experiments where we train models using sampling schedules obtained from running AutoSampling on various model structures, and the results demonstrate the transfer ability of the optimal sampling distribution among models, as shown in Table.5.

---

### Official Review · AnonReviewer4 · 2020-11-02
**a work that explores data-sampling scheme in PDT framework**

**Rating:** 5
**Confidence:** 4

**Review:**

This work presents an interesting exploration of learning optimal data sampling probability. It is formulated with a population based training strategy. To this end, authors propose to record the data frequencies through all previous steps and thereby generate the sampling policy for the next step.

It has been a recent trend to make almost all key factors of deep networks learnable, such as differentiable data augmentation. This work is taking a novel aspect for effectively training neural networks, since rare effort has been devoted to make data sampling learnable. Authors frame the task in PBT and take incremental modification to PBT to adapt to the new problem. I have no complaint regarding the task per se. However, the empirical validation seems finished in rush and not sufficient.

The chosen baselines are essentially standard sampling scheme or variants of the proposed method. Authors should compare with a few state-of-the-art data-sampling or data-reweighting methods, such as Focal Loss proposed by He et al. In addition, Tables 2 and 3 should be clarified in the rebuttal. It seems that very small margins are observed in comparison with random exploration. The follow-up analysis attributes the success to the emphasis of hard examples during training, which is consistent to previous common practice. However, authors are encouraged to go deeper for the analysis, beyond the selected illustrative samples in Figure 3. For example, it may be meaningful to check the alignment of data weights and classification scores.

---

> ### Author Response · Authors · 2020-11-23
> **Response to Reviewer 4**
>
> We thank the reviewer for the valuable feedback and would like to address some of the reviewer’s concerns.
>
> * "The chosen baselines are essentially standard sampling scheme or variants of the proposed method. Authors should compare with a few state-of-the-art data-sampling or data-reweighting methods, such as Focal Loss proposed by He et al."
>
>     * We wish to demonstrate that the proposed method is able to exceed the performance of state-of-the-art baselines no matter what techniques (knowledge distillation, data resampling, loss reweighting) they utilize. For example, as shown in Table.1 our methods outperforms [1] by a clear margin.
>     *Moreover, about the comparison with top data-resampling methods, in A.2 of the appendix we include comparison between the proposed methods and recent state-of-the-art methods [2,3] and show better performance in their respective settings, as shown in Table.6.
>     *We also ran experiments using focal loss, but the performance on regular classification datasets like ImageNet is no obliviously better than random sampling baselines. We try to list and compare with related works as much as possible and aim to demonstrate that the proposed AutoSampling can achieve top performances.
>
> * "Tables 2 and 3 should be clarified in the rebuttal. It seems that very small margins are observed in comparison with random exploration. "
>     * The AutoSampling with the random exploration is allowing the sampler to learn different sampling frequencies for each training image. The mixture exploration directs the learning process away from local minimum by mixing uniform distribution into the sampling distribution, which allows a more thorough search. CIFAR-10/100 datasets are relatively small in size and complexity, and we think it is easier to reach a robust sampling schedule even with just the random exploration. Therefore the mixture exploration does not outperform by a large margin upon an already strong sampling schedule. However, as shown in Table.3, the mixture exploration outperforms the random exploration by a clear of 0.8 margin on ImageNet that is complex and large enough to see the full potential of the mixture exploration.
>
> * "The follow-up analysis attributes the success to the emphasis of hard examples during training, which is consistent to previous common practice. However, authors are encouraged to go deeper for the analysis, beyond the selected illustrative samples in Figure 3. For example, it may be meaningful to check the alignment of data weights and classification scores."
>     * This is a great question. We wish to clarify that in the analysis of Section 4.3 we mean to illustrate one characteristic of the sampling schedule learned by AutoSampling, that is, the positive correlation between sampling frequency and visual quality. But this is not the only characteristic of the learned schedule. As advised by the reviewer, we check the alignment between the sampling frequency and the loss values of each data, and added discussion and illustration in A.3 of the Appendix. The analysis shows that the learned schedule is not strongly correlated to the loss values of data, which indicates AutoSampling learns a schedule beyond simply mining visually hard examples or examples deemed difficult by loss values.
>
> Reference
>
> 1. Xiao Jin, Baoyun Peng, Yichao Wu, Yu Liu, Jiaheng Liu, Ding Liang, Junjie Yan, and Xiaolin Hu. Knowledge distillation via route constrained optimization. In The IEEE International Conference on Computer Vision (ICCV), October 2019.
>
> 2. Tyler B Johnson and Carlos Guestrin. Training deep models faster with robust, approximate im- portance sampling. In S. Bengio, H. Wallach, H. Larochelle, K. Grauman, N. Cesa-Bianchi, and R. Garnett (eds.), Advances in Neural Information Processing Systems 31, pp. 7265–7275. Curran Associates, Inc., 2018. URL http://papers.nips.cc/paper/7957.
>
> 3. Angelos Katharopoulos and Franc ̧ois Fleuret. Not all samples are created equal: Deep learning with importance sampling. CoRR, abs/1803.00942, 2018. URL http://arxiv.org/abs/1803. 00942.

---

### Decision · Program_Chairs · 2021-01-07
**Final Decision**

**Decision:**

Reject

**Comment:**

The work focuses on a new method for sampling hyper-parameter based on an "Population-Based Training" schedule that tend to sample more often configurations that gave good performances in the past. The authors have conducted experiments to verify the superior of their method, especially for the effectiveness and generalisability.

Pros:
- simple method that can be implemented without much effort,
- good empirical performances on Imagenet,
- paper well organised and written.

Cons:
- lack of explanation about the DensNet121 performance degradation [partially addressed in the rebuttal],
- additional simple experiments in Section 4.4 were recommended to evaluate the generality of the method [addressed in Table 5],
- empirical validation seems not sufficient [partially addressed in the rebuttal],
- similarity with respect to prior art, such as the focal loss [partially addressed in the rebuttal],
- clarification of the randomisation strategy in experiments [addressed in the rebuttal].

Despite most of the issues being addressed, the reviewers decided that this paper would benefit more work to be accepted for the conference this year.